# Prevalence and determinants of medication adherence among hypertensive patients: An institution-based cross-sectional study

Azaz Bin Sharif [1,2]*, Syed Sharaf Ahmed Chowdhury[1], Md. Zakir Hossain[3], Md. Anwar Hossain[3], Ahmed Hossain[4], Hasan Mahmud Reza[5]

1 Global Health Institute, North South University, Dhaka, Bangladesh, 2 Department of Public Health, North South University, Dhaka, Bangladesh, 3 Hypertension and Research Center, Rangpur, Bangladesh, 4 College of Health Sciences, University of Sharjah, Sharjah, United Arab Emirates, 5 Department of Pharmaceutical Sciences, North South University, Dhaka, Bangladesh

* azaz.sharif@northsouth.edu

## Abstract

### Background

Adherence to antihypertensive medication is a key management strategy for hypertension, which gives rise to the necessity to get a clear picture of medication adherence among patients with hypertension in Bangladesh. This study aims to determine the prevalence of antihypertensive medication adherence and its associated factors among patients with hypertension receiving treatments at a Hypertension Center in Bangladesh.

### Methods

An institution-based cross-sectional study was conducted on 352 adult patients with hypertension registered in the Rangpur Hypertension and Research Center. The outcome variable for the study was medication adherence to the antihypertensive drug, which was measured using the Morisky Medication Adherence Scale (MMAS-8). Descriptive analysis was conducted to show the distribution of the study participants, Pearson's chi-square test was applied to explore associations between categorical response and explanatory variables, and bivariable and multivariable logistic regression analyses were conducted to explore factors associated with medication adherence. All statistical analyses were conducted using Stata version 17.0.

### Results

The prevalence of good medication adherence among patients with hypertension in the HRC, R was 54.83%. Among various reasons for taking the medicine irregularly, forgetfulness (20.29%) was the most common cause reported by the participants, followed by a busy work schedule (7.71%). Among the study participants, married patients were found

Data availability statement: Data can be downloaded from the following URL: https://figshare.com/s/f272e36bcae006ff6398?file=49774473.

Funding: The North South University's Conference and Travel Related Grant (CTRG) (Grant No: CTRG-23_SHSL-41) supported this work; the grant recipient was Dr. Azaz Bin Sharif.

Competing interests: The authors declare that they have no competing interest.

to have higher medicine adherence (AOR= 3.81; 95% CI: 1.34–10.89) than unmarried patients. Compared to patients who had hypertension for less than or equal to 5 years, patients suffering from hypertension for 6–10 years had 68% (AOR= 0.32; 95% CI: 0.11–0.95) and patients suffering for more than 10 years had 72% (AOR= 0.28; 95% CI: 0.09–0.84) lower odds of medicine adherence, respectively. Patients who were diagnosed by non-professionals had 81% (AOR= 0.19; 95% CI: 0.06–0.61) lower odds of medicine adherence compared to the patients who were diagnosed by health professionals.

## Conclusion

This study observed low medication adherence among patients with hypertension, where forgetfulness and a busy work schedule were reported to be the primary reasons for such non-adherence. Patients who are not married, who have been suffering for a long, and who have not been diagnosed by a health professional manifested as significant influencing factors for non-adherence.

## Introduction

Cardiovascular diseases (CVDs) are one of the major non-communicable diseases worldwide. It is recognized as the world's biggest killer, considering the 13% contribution to the overall cause of death around the world since 2000 [1]. Hypertension is a major cardiovascular disease risk factor and a significant global health concern [2,3]. It affects around 1.28 billion adults globally, two-fifths of which belong to low-and-middle-income countries (LMIC) [4]. However, hypertension's asymptomatic nature makes it harder to diagnose and treat. Nearly half of the population (46%) in the world is unaware of their hypertensive status [5], and it is believed that 42% of patients with hypertension who are diagnosed as hypertensive get treated, and only 21% have good control [4]. Hypertension plays a significant role in developing non-communicable diseases (NCDs) and is also a disease. Alongside causing deadly physical problems, it contributes to a considerable economic burden [6]. Regardless of the increasing prevalence, awareness and treatment seeking are still low, especially in low- and middle-income countries like Bangladesh [7–9].

The prevalence of hypertension has been on the rise over the past decade [6]. Rapid urbanization and over-digitalization have resulted in an increased sedentary lifestyle and dietary shift from traditional to fast food [10,11]. This, in turn, has grown to be one of the major contributors to the rapid epidemiological transition from communicable to non-communicable disease [10]. From 25.7% in 2011, the prevalence of hypertension among Bangladeshi adults jumped to 39.4% in 2017–18 [12]. This scenario is not only causing serious health problems but also giving rise to a significant economic burden [13]. In a systematic review, the overall weighted pooled prevalence of hypertension was found to be 20% in Bangladesh [10]. Considering the devastating consequences of hypertension, the World Health Organization (WHO) aims to reduce hypertension prevalence by 33% by 2030 [4].

Antihypertensive medication is a key management strategy for hypertension. However, medication non-adherence is one of the main reasons for not achieving optimal blood pressure (BP) control [14]. A meta-analysis found that patients adherent to antihypertensive medications were 3.44 times more likely to have good blood pressure control than non-adherent patients [15]. Despite the proven efficacy of pharmacologic treatment, medication adherence among patients with hypertension is often suboptimal [2,3]. This lack of adherence to treatment regimens can lead to poor blood pressure control and an increased risk of adverse health outcomes [16]. Several factors have been identified to be influencing medication adherence. These include socio-economic factors such as age, civil status, educational attainment, and employment status; patient-related factors such as health literacy, awareness, knowledge about hypertension, attitude towards the disease, self-efficacy, and social support, therapy-related factors such as inconsistent drug regimen schedule and use of alternative medicines; and condition-related factors such as illness perception and presence of comorbidities [3].

A study conducted in Riyadh, Saudi Arabia found that only 42.2% of patients with hypertension were adherent to their antihypertensive medications [2]. The study also revealed that patients with comorbidities and those on multiple medications were at a higher risk of medication non-adherence [2]. Similarly, a systematic review conducted in the Philippines reported that medication adherence rates are as low as 66% among patients with hypertension [3]. Bangladesh is no exception in this regard. A study conducted in Narail, Bangladesh, observed that medication adherence of antihypertensive medicines was less than half (46.9%) [17]. A systematic review conducted in developing countries, including Bangladesh, revealed that the prevalence of medication adherence among hypertensive patient is around 47.34% [18]. Currently, 50–70% of Bangladeshi patients have uncontrolled hypertension, largely due to poor medication adherence [19]. A study in Matlab indicates that only 26% of the patients with hypertension have good BP control [19].

It's expected that the trend of hypertension will continue to rise due to the demographic transition towards older age groups and an increase in overweight and obesity among the population of Bangladesh [20]. Poor adherence to medicine worsens hypertension and leads to other cardiovascular diseases. Proper use of antihypertensive medication is crucial for managing hypertension and reducing cardiovascular disease risk, which gives rise to the necessity of getting a clear picture of medication adherence among patients with hypertension in Bangladesh. Although in Bangladesh, medication adherence among patients with hypertension has been measured in both community and hospital settings, all of them cover either urban or rural communities. To our knowledge, till now, no studies have been conducted in a semi-urban setting covering both urban and rural populations. Therefore, this study aims to determine the prevalence of antihypertensive medication adherence and its associated factors among patients with hypertension receiving treatments at the Hypertension and Research Center, Rangpur (HRC, R), Bangladesh.

## Methodology

### Inclusion criteria

The study participants were selected using the following criteria

1. Must be registered at the Hypertension and Research Center, Ranpur for 6 months

2. Receive the treatment for hypertension from this center only

3. Must have a basic understanding of the native language (Bangla) and be able to answer questions independently

4. Must not be critically ill during the interview period

### Study design and setting

An institution-based cross-sectional study was conducted at the Rangpur Hypertension and Research Center. Based on the inclusion criteria, the adult participants who were registered in the Hypertension and Research Center, Rangpur for at least six months before the data collection were included in the study.

## Ethical approval

Ethical clearance was obtained from the Institutional Review Board (IRB) of North South University (ethical approval reference number: 2023/OR-NSU/IRB/0408). The study's objectives were provided to each respondent before data collection, and informed consent was obtained from each respondent. We also ensured that anonymity and confidentiality were maintained. The moral principles were set down in the 1964 Declaration of Helsinki and later changes were followed. An informed written consent was provided to each participant before the study.

## Sample size calculation

We estimated the sample size for the study using the proportion formula,

$$n = \frac{z^2 p(1-p)}{d^2} \qquad (1)$$

where n is the expected sample size, z is the test statistic value corresponding to the 95% level of confidence, which is 1.96, and p is the prevalence of medication adherence among the patients with hypertension, which was 73.8% considered from the previous study conducted in Bangladesh [19], and d is the margin of error, which was considered 5% for this study. The estimated sample size was 292 (rounded). However, we collected a sample of 352.

## Sampling technique

The participants for this study were selected by systematic random sampling. For this, we observed the average number of registered patients visiting the facility daily from the record book with appropriate permission. Over the last six months, 20 registered patients visited the facility daily. Considering the time frame for data collection, we calculated the interval for collecting the required number of samples to be 5. Therefore, every 5th participant visiting the facility for consultation between June 2023 and September 2023 was interviewed.

## Data collection

A pre-tested semi-structured questionnaire was used, where the questions were selected and organized based on a thorough literature review [21–26]. Trained data collectors administered the questionnaire, and pretesting was done among 30 participants, randomly selected from the registered hypertensive patient of HRC, R, who were later excluded from the final sample. The questionnaire gathered information on the participants' sociodemographic characteristics, information about their hypertension status, family history, and chronic disease status, and the medication adherence of the participants was measured using an 8-item Morisky Medication Adherence Scale (MMAS-8). We have obtained permission to use the MMAS-8 scale from the respective authority (Certificate number: 7115-6064-9595-2723-8695).

## Outcome variable

The outcome variable for the study was the medication adherence to antihypertensive drug among hypertensive male and female adults aged 18 and above registered in the HRC, R. Medication adherence was measured by the Morisky Medication Adherence Scale (MMAS-8). The scale consists of 8 questions each with a binary response, 'yes' and 'no'. The response was coded as 0 for no and 1 for yes [27–29]. This MMAS-8 scale was tested on a sample of patients with hypertension by Morisky et. al. [27–29] and was found to have good sensitivity and predictive validity. The 8-item scale was found to be 93% sensitive and 53% specific on that test [29]. This is a widely used scale to determine adherence to the antihypertensive medicine regimen among patients with hypertension. Medication adherence is calculated by adding the response to all the questions. The results for the original MMAS-8 scale are categorized as high, medium, and low adherence to the antihypertensive medication for the score of 8, 6 to <8, and <6 respectively [27–29]. A binary outcome was

created for this study by using 6 as a cutoff point and categorized as poor if the score was 6 or lower [29]. The estimated sensitivity and specificity of the scale for the binary categorization were found to be the same as the original one.

### Independent variables

Based on the review of the previous literature, Independent variables considered in this study were: age (<45 years, 45–64 years, 65years and above), educational status of the participants (up to primary, secondary, and higher), employment status of the participants (working and not working), marital status (married and unmarried), family's monthly income (categorized as 20,000 or less BDT/month and >20,000), HTN diagnosis personnel (health professional, non-health professional), duration of hypertension diagnosis(≤5 years, 6–10 years and >10 years),. The response for the variables; family history of hypertension and medication taken for any other chronic diseases was binary (yes, no).

### Statistical analysis

The distribution of the independent variables among the study participants was shown as a part of the descriptive analysis. Differences in the medication adherence status among the patients with hypertension by their sociodemographic characteristics, hypertensive history, family history, and chronic disease history were observed using Pearson's chi-square test. Before conducting regression analysis, a test for multi-collinearity was conducted to confirm whether there are any correlated independent variables. The VIF didn't demonstrate any significant collinearity among the independent variables. Both the bivariable and multivariable logistic regression analyses were conducted and all the uncorrelated variables were included in the multivariable model. Both the crude and adjusted odds ratios along with their corresponding 95% confidence interval were reported. The P value less than 0.05 was considered statistically significant. All analysis was done in the STATA v.17.0

## Results

### Sociodemographic and lifestyle characteristics

In our study, the participant's ages ranged from 24 years to 98 years with a mean of 53.29 and a standard deviation of 11.64. The highest number of participants belonged to the 45–64 years age group (61.65%) (Table 1). Among the study participants, only 34.66% of them were found to have completed higher secondary education and about 34.38% were found to have completed secondary education. Respondents were mostly married (88.92%). About half of the participants (50.85%) were reported to be not working. The highest proportion of the participants (69.32%) was observed to be in the income group of 20,000 or less BDT per month. More than half of the study participants (61.36%) were overweight. Family history of hypertension (father's family) was reported by about 65.91% of the study participants. About 44.32% of the participants were observed to be diagnosed with hypertension more than 10 years ago. Around 85.51% were diagnosed with HTN by health professionals and almost half of the participants (45.74%) had a history of taking medication for chronic diseases.

### Prevalence and distribution of medication adherence

The prevalence of good adherence to the antihypertensive medicine among 352 patients with hypertension in the HRC, R was 54.83%. Participants belonging to the age group of <45 years were found to have a higher prevalence of good medication adherence (59.15%). From Table 2, it is evident that there is no significant difference in the prevalence of good adherence among the participants with different levels of educational attainments. However good medication adherence was found highest among the participants completing their higher secondary education (57.38%). A significantly (p = 0.012) higher percentage of good medication adherence was observed among the married participants (57.19%) compared to their counterparts (35.90%). Higher prevalence of good antihypertensive medication adherence was found

**Table 1. Distribution of Sociodemographic and Lifestyle Characters of the Participants.**

| Variables | | Frequency (n) | Percentage (%) |
|---|---|---|---|
| Age | <45 years | 71 | 20.17 |
| | 45-64 years | 217 | 61.65 |
| | 65 years and above | 64 | 18.18 |
| Highest level of education | Upto primary | 121 | 34.38 |
| | Secondary | 109 | 30.97 |
| | Higher secondary | 122 | 34.66 |
| Marital status | Unmarried | 39 | 11.08 |
| | Married | 313 | 88.92 |
| Occupation | Working | 173 | 49.15 |
| | Not working | 179 | 50.85 |
| Family monthly income (in BDT) | 20000 or less | 244 | 69.32 |
| | >20000 | 108 | 30.68 |
| BMI | Normal weight | 136 | 38.64 |
| | Overweight | 216 | 61.36 |
| Duration of hypertension diagnosis | ≤5 years | 85 | 24.15 |
| | 6-10 years | 111 | 31.53 |
| | >10 years | 156 | 44.32 |
| HTN diagnosis personnel | Health professional | 301 | 85.51 |
| | Non-health professional | 51 | 14.49 |
| Family history of HTN | Yes | 232 | 65.91 |
| | No | 120 | 34.09 |
| Take any medication for chronic diseases | Yes | 161 | 45.74 |
| | No | 9 | 2.96 |

(60.19%) among the participants belonging to the lower-income group. Participants with normal BMI (61.03%) and a family history of hypertension (56.47%) were found to have better adherence to antihypertensive medicine than their corresponding counterparts. Good medication adherence was found to be higher among the participants who were diagnosed with hypertension for 5 years or less (60%) as shown in Table 2. Respondents diagnosed by health professionals and those who were on medication for chronic diseases showed better medication adherence at 56.48% and 59.63% respectively compared to their counterparts (Table 2).

### Reason for non-adherence

Fig 1 shows the self-reported reasons for being non-adherent to antihypertensive medicine among the study participants in the Hypertension and Research Center, Rangpur. Among various reasons for taking the medicine irregularly, forgetfulness (20.29%) was the most common cause reported by the participants followed by a busy work schedule (7.71%), evident from Fig 1. Around 4.29% of the patients mentioned some other reasons (fear of side effects, traveling, and boredom) for not taking the antihypertensive medicine regularly (Fig 1).

### Factors associated with medication adherence

Table 3 shows the result of multivariable logistic regression analysis. Among the study participants, married participants were found to have 3.81 times higher odds (95% CI: 1.34–10.83) of being adherent to the antihypertensive medicine compared to the unmarried participants. Duration of hypertension diagnosis was found to be significantly associated with good medication adherence among the participants in the Rangpur Hypertension and Research Center. It is evident from

**Table 2. Prevalence and Distribution of Medication adherence among Participants.**

| Variables | | Poor Adherence 45.17% | Good Adherence 54.83% | P value |
|---|---|---|---|---|
| | | Frequency (%) | Frequency (%) | |
| Age (in years) | <45 years | 29 (40.85) | 42 (59.15) | 0.541 |
| | 45-64 years | 103 (47.47) | 114 (52.53) | |
| | 65 years and above | 27 (42.19) | 37 (57.81) | |
| Highest level of education | Upto Primary | 53 (43.80) | 68 (56.20) | 0.535 |
| | Secondary | 54 (49.54) | 55 (50.46) | |
| | Higher secondary | 52 (42.62) | 70 (57.38) | |
| Marital status | Unmarried | 25 (64.10) | 14 (35.90) | 0.012 |
| | Married | 134 (42.81) | 179 (57.19) | |
| Occupation | Working | 72 (41.62) | 101 (58.38) | 0.188 |
| | Not working | 87 (48.60) | 92 (51.40) | |
| Family monthly income (in BDT) | 20000 or less | 116 (47.54) | 128 (52.46) | 0.179 |
| | >20000 | 43 (39.81) | 65 (60.19) | |
| BMI | Normal weight | 53 (38.97) | 83 (61.03) | 0.064 |
| | Overweight | 106 (49.07) | 110 (50.93) | |
| Family History of Hypertension | Yes | 101 (43.53) | 131 (56.47) | 0.391 |
| | No | 58 (48.33) | 62 (51.67) | |
| Duration of Hypertension Diagnosis | ≤5 years | 34 (40.00) | 51 (60.00) | 0.248 |
| | 6-10 years | 57 (51.35) | 54 (48.65) | |
| | >10 years | 68 (43.59) | 88 (56.41) | |
| HTN diagnosis personnel | Health professional | 131(43.52) | 170(56.48) | 0.131 |
| | Non- Health professional | 28(54.90) | 23(45.10) | |
| Take any medication for chronic diseases | Yes | 65(40.37) | 96(59.63) | 0.386 |
| | No | 5(55.56) | 4(44.44) | |

Table 3 that, with the increase in the duration of hypertension diagnosis the odds of good medication adherence of the participants were found to decrease significantly with 0.32 and 0.28 odds for 6–10 years and >10 years duration, respectively (95% CIs: 0.11–0.95 for the former, and 0.09–0.84 for the latter). The study found that participants who were initially diagnosed by non-health professionals had lower odds of good medication adherence (AOR= 0.19; 95% CI: 0.06–0.61) compared to those who were diagnosed by the health professionals (Table 3).

## Discussion

The current study aims to identify the prevalence of medication adherence among patients with hypertension in a tertiary center in Rangpur district in Bangladesh. Prevalence of good adherence to antihypertensive medicine was found to be higher among the study participants. After adjusting for all the variables, marital status, duration of hypertension diagnosis, and diagnosis personnel of hypertension were found to be significantly associated with medication adherence among the patients with hypertension.

The overall prevalence of good adherence to antihypertensive medicine among the participants with hypertension in the Hypertension and Research Center Rangpur was found to be 54.83%. This finding is in line with the previous studies conducted in Bangladesh [30], Ethiopia [31], and Malaysia [32]. The plausible reason behind this finding could be the high level of awareness among patients with hypertension due to improved healthcare facilities [30]. The selection of the study site being a tertiary center for hypertensive treatment could be another reason behind the increased prevalence of good

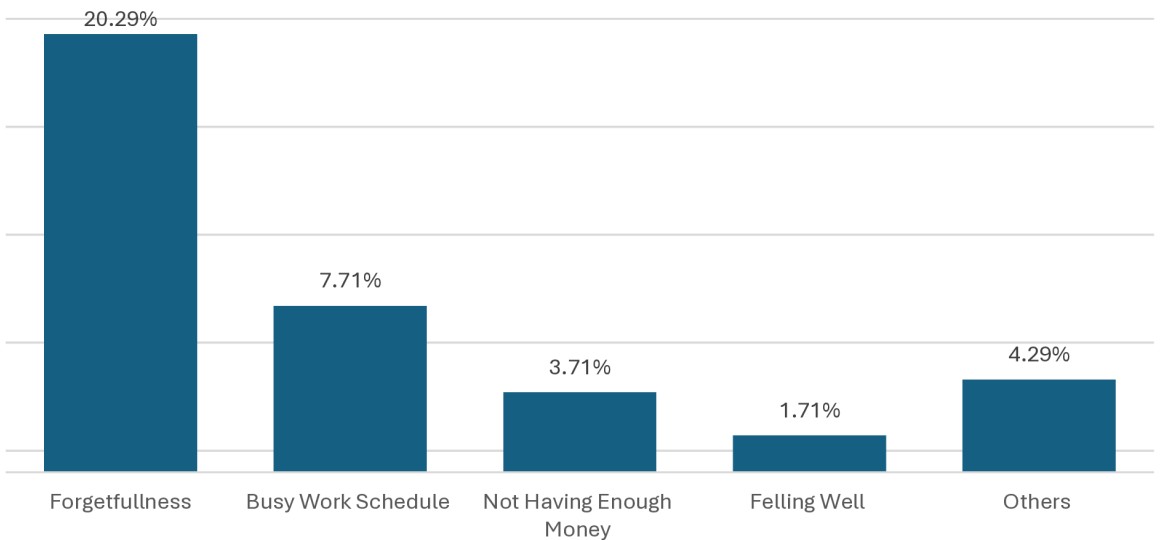

**Fig 1. Self-reported reasons for not taking antihypertensive medicine regularly by patients with hypertension.**

**Table 3. Factors Associated with Medication adherence.**

| Variables | | COR | 95% CI | AOR | 95% CI |
|---|---|---|---|---|---|
| Age (in years) | <45 years | 1.0 (Ref.) | | 1.0 (Ref.) | |
| | 45-64 years | 0.76 | 0.44, 1.32 | 0.44 | 0.16, 1.21 |
| | 65 years and above | 0.95 | 0.48, 1.88 | 0.65 | 0.18,2.33 |
| Highest level of education | Upto Primary | 1.0(Ref.) | | 1.0 (Ref.) | |
| | Secondary | 0.80 | 0.47, 1.33 | 0.38 | 0.16, 0.92 |
| | Higher Secondary | 1.05 | 0.63,1.74 | 0.41 | 0.15, 1.13 |
| Marital status | Unmarried | 1.0 (Ref.) | | 1.0 (Ref.) | |
| | Married | 2.39 | 1.19, 4.76 | 3.81 | **1.34, 10.83** |
| Occupation | Working | 1.0 (Ref.) | | 1.0 (Ref.) | |
| | Not working | 0.75 | 0.49, 1.15 | 0.71 | 0.31, 1.62 |
| Family monthly income (in BDT) | 20000 or less | 1.0 (Ref.) | | 1.0 (Ref.) | |
| | >20000 | 1.37 | 0.86, 2.17 | 1.10 | 0.49, 2.47 |
| BMI | Normal weight | 1.0 (Ref.) | | 1.0 (Ref.) | |
| | Overweight | 0.66 | 0.43, 1.02 | 0.54 | 0.25, 1.14 |
| How many years ago were you first diagnosed as hypertensive (in years) | ≤5 years | 1.0 (Ref.) | | 1.0 (Ref.) | |
| | 6-10 years | 0.63 | 0.36, 1.12 | 0.32 | **0.11, 0.95** |
| | >10 years | 0.86 | 0.50, 1.48 | 0.28 | **0.09, 0.84** |
| HTN diagnosis personnel | Health professional | 1.0(Ref) | | 1.0 (Ref.) | |
| | Non-health professional | 0.63 | 0.35, 1.15 | 0.19 | **0.06, 0.61** |
| Family history of HTN | Yes | 1.0 (Ref.) | | 1.0 (Ref.) | |
| | No | 0.83 | .53,1.28 | 0.80 | 0.37, 1.72 |
| Take any medication for chronic diseases | Yes | 1.0(Ref) | | 1.0(Ref) | |
| | No | 0.54 | .14, 2.09 | 0.77 | 0.17, 3.49 |

adherence among the hypertensive participants. On the contrary, 45.17% of the study participants were found to have poor adherence to medicine. A higher prevalence of poor adherence was found in other studies conducted in Bangladesh [23], China [33–35], Peru [36], and Congo [37] whereas a lower prevalence of poor adherence to antihypertensive medicine was observed in separate several studies conducted in Bangladesh [19,24], Ethiopia [31], and Hongkong [22]. While several factors like population, study design, and study setting could explain the difference in the prevalence of poor adherence in different studies, cultural beliefs and socio-demographic differences could be possible reasons behind the observed variation [21,22].

After adjustment for different sociodemographic variables, the marital status of the hypertensive participants in the Rangpur Hypertension and Research Center was found to be significantly associated with antihypertensive medication adherence. This study's findings coincide with the previous study findings conducted in Bangladesh [26] and Nepal [38]. This finding could be explained by the additional support received by the married patient from their spouses, which helps them to stick to the medication and results in greater medication adherence [39–42].

Duration of Hypertension diagnosis was associated with good medication adherence among the participants. With the increase in the duration of hypertension diagnosis medication adherence was found to be poor. A similar result was observed in a previous Bangladeshi [26] and Nepali population [38]. The probable reason behind this finding could be the increased year of diagnosis leads to increased use of medicine which could be perceived as a burden or barrier to becoming adherent to medicine. Besides, increased duration of medicine use could be associated with increased chances of side effects related to hypertensive medicine, which could be another possible reason for not being adherent to antihypertensive medicine among the participants.

Good medication adherence was found to be low among the participants who were diagnosed by non-health professionals. The plausible reason behind this finding could be participants who were diagnosed by non-health professionals didn't receive enough information on the consequences of non-adherence to antihypertensive medicine. The lack of proper knowledge about the advantages of medication adherence in treating hypertension and how medicine lowers BP could be a probable reason behind the non-adherence among the participants who were diagnosed by non-health professionals [43].

In this study, we observed that hypertensive participants with higher BMI had poor medicine compliance compared to the participants with normal BMI. Although not found statistically significant, this finding was found in line with the previous study [26]. It was also observed from the study that medication adherence was not found to be associated with the age, educational level, occupation, or monthly family income of the participants.

## Limitations

The study has some limitations. Firstly, we conducted a single-centered study that limited the generalizability of our results. Secondly, data were being collected cross-sectionally, which does not provide information on exposure or outcome precedence. Therefore, we couldn't establish a causal relationship between medication adherence and other independent variables. Lastly, self-reporting responses to medication adherence could give rise to recall bias, given the past medication history was recorded from the patients.

## Conclusion

This study observed low medication adherence among patients with hypertension, where forgetfulness and a busy work schedule were reported to be the major reasons for such non-adherence. Patients who are not married, who have been suffering for a long, and who have not been diagnosed by a health professional manifested as significant influencing factors for non-adherence. Counseling and strategies, such as the use of pill boxes and electronic reminders, may help increase medication adherence.

## Acknowledgment

We place our heartiest gratitude to Hypertension and Research Center, Rangpur Bangladesh for their continuous support throughout the study period. We are also grateful to North South University for their logistic Support

## Author contributions

**Conceptualization:** Azaz Bin Sharif, Syed Sharaf Ahmed Chowdhury.

**Data curation:** Syed Sharaf Ahmed Chowdhury.

**Formal analysis:** Syed Sharaf Ahmed Chowdhury.

**Methodology:** Azaz Bin Sharif, Syed Sharaf Ahmed Chowdhury, Ahmed Hossain, Hasan Mahmud Reza.

**Supervision:** Azaz Bin Sharif, Md. Zakir Hossain, Ahmed Hossain, Hasan Mahmud Reza.

**Writing – original draft:** Azaz Bin Sharif, Syed Sharaf Ahmed Chowdhury, Md. Anwar Hossain.

**Writing – review & editing:** Azaz Bin Sharif, Md. Zakir Hossain, Ahmed Hossain, Hasan Mahmud Reza.

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
