## [Decision Letter · Decision Letter 0]

23 Jul 2024

PONE-D-24-18421Hypertension and Medication Adherence: An Institution-based Study at Hypertension and Research Center, RangpurPLOS ONE

Dear Dr. Sharif,

Thank you for submitting your manuscript to PLOS ONE. After careful consideration, we feel that it has merit but does not fully meet PLOS ONE’s publication criteria as it currently stands. Therefore, we invite you to submit a revised version of the manuscript that addresses the points raised during the review process.

We look forward to receiving your revised manuscript.

Kind regards,

Eyob Alemayehu Gebreyohannes, PhD

Academic Editor

PLOS ONE

Journal Requirements:

"The Study received funding from the North South University as a part of the Conference and Travel Related Grant (CTRG)."

"We place our heartiest gratitude to Hypertension and Research Center, Rangpur Bangladesh for their continuous support throughout the study period. We are also grateful to North South University for their logistic and Financial Support"

Please note that funding information should not appear in the Acknowledgments section or other areas of your manuscript. We will only publish funding information present in the Funding Statement section of the online submission form. Please remove any funding-related text from the manuscript. 

5. In this instance it seems there may be acceptable restrictions in place that prevent the public sharing of your minimal data. However, in line with our goal of ensuring long-term data availability to all interested researchers, PLOS’ Data Policy states that authors cannot be the sole named individuals responsible for ensuring data access (http://journals.plos.org/plosone/s/data-availability#loc-acceptable-data-sharing-methods).

Reviewers' comments:

Reviewer's Responses to Questions

**Comments to the Author**

1. Is the manuscript technically sound, and do the data support the conclusions?

Reviewer #1: Yes

Reviewer #2: Yes

2. Has the statistical analysis been performed appropriately and rigorously? 

Reviewer #1: Yes

Reviewer #2: I Don't Know

3. Have the authors made all data underlying the findings in their manuscript fully available?

Reviewer #1: No

Reviewer #2: No

4. Is the manuscript presented in an intelligible fashion and written in standard English?

Reviewer #1: No

Reviewer #2: Yes

5. Review Comments to the Author

Reviewer #1: Title: Hypertension and Medication Adherence: An Institution-based Study at Hypertension and Research Centre, Rangpur

The authors came with a good research area that investigates the prevalence of medication adherence and associated factors among patients with hypertension who received follow-up care at the Hypertension and Research Centre in Rangpur, Bangladesh.

However, despite it has a potential, the manuscript is not sufficient for publication in its current form and it needs a careful revision.

Minor comments

- The word/phrase “hypertension/hypertensive patients” is not a good usage of language because it focuses on labelling of patients. Despite you may find it in a lot of literature, but it has a negative and discriminative sense of language and not advised to use. “Patients with hypertension” is generally preferred in scientific writing. It focuses on the condition rather than labelling the patient. Please you need to change it in all sections of the manuscript.

- The MMAS-8 likely requires permission to use, right? "Have you obtained permission to use the MMAS-8 in your study? It's important to ensure you have the necessary permissions before publishing your findings. The authors need to ensure this. Other ways you are making yourselves and others busy with nothing.

- The journal requires a manuscript fully available without restriction, but you put restriction to your data availability. Please refer the Data availability policy of the journal.

Major comments

Title

#1.... The title as it stands is ambiguous, and it needs to be clarify with your interest to investigate. Determining the prevalence of medication adherence and its associated factors in patients with hypertension is the aim of the study you are going to investigate, right? So, please could you make it clear in this context?

Abstract

#1… Is Pearson’s chi-square test used to explore factors associated with medication adherence?

#2… bivariate and multivariable logistic regression analyses: is it bivariable or bivariate?

#3…It would be good if you could include statistical analysis software and your declarations used for this particular study.

#4…Result or results

#5… “Duration of hypertension diagnosis (AOR: 0.32; 95% CI: 0.11-0.95 for hypertension diagnosed <10 years back and AOR: 0.28; 95% CI: 0.09-0.84 for >10 years)”

I didn’t find the point on these results, or if there is something not clearly presented, please could you clarify it? You mentioned both < 10 and > 10 years duration of hypertension diagnosis as a factor for medication adherence. What is the counterpart reference here? I mean, patients living < 10 and >10 years with hypertension had poor/good medication adherence than what?

#6… Better to use “AOR =” instead of “AOR:”

#7…Your conclusion is not aligned with your main outcome variable. Your main outcome variable is prevalence of medication adherence, right? If so, your conclusion should inform readers what does the outcome implies. These things mentioned here might be used for your justification and recommendation about the outcome.

Introduction

#1… organized and written well, but some evidences are mentioned without a citation, please you need to cite and acknowledge appropriate literature for it.

#2… there is a redundancy of the word hypertension or the hypertension in the first paragraph, better to re-write without redundant it.

#3… Make sure of using updated evidence, while you are mentioning the WHO and other international organizations’ reports and evidence. For instance, the WHO published an update on 16 March 2023. https://www.who.int/news-room/fact-sheets/detail/hypertension#:~:text=Approximately%201%20in%205%20adults,33%25%20between%202010%20and%202030.

Methods

#1… “..Adult participants registered...”… What does registered means? Did they receive antihypertensive medications? If so, why we couldn’t mention them …adult participants who received antihypertensive medications?”

#2… Why you chose six months? Is there any implication or rationale? Why not three months?

#3... You said participants were included based on the inclusion criteria. You should better to clarify these inclusion criteria more than this. You need to clarify our confusion in the abstract section above here.

4#... What is 5th? You did not inform us how you did the sample size calculation, and you just wrote only “the sampling interval-K”. Please move this section including sampling technique you used after the sample size calculation because you are going to calculate k after determined the sample size.

#5…Why you would like to use a larger sample size of 325, while you found a sample size of ~292 is enough for this stud?

#6… You used a thorough literature review to organize the questionnaire for this study. You need to cite and acknowledge these literature used for developing your questionnaire.

#7. Pearson’s chi-square test here in the method section is presented appropriately unlike in the abstract.

#8… bivariate, the same as state in the abstract

#9…better to incorporate your declaration of statistical significant p-value here.

Results

#1…When I just observed the number of respondents [352] in the table 1, I was surprised to observe a 100% response rate. This is rare in many studies. I wander if you could include this with some methods used to achieve a 100% response rate. Your experience may be valuable to other researchers.

#2… Table 1 and its description: you described using texts almost all of the variables in the table 1. Please put only some texts to describe tables, other ways it looks duplication or no need of tables.

#3... The same is true for Table 2 and its description texts. I think the need of table is to present variables in short, so focus on the main variable you would like to describe using texts and leave others in the table with reference [table citation] for the readers.

#4… Again you mentioned all variable using texts to figure 1, and then you are presenting as a figure again. Please avoid the figure and present only using texts or if you want to keep both mentioned the text like this: “Forgetfulness (20.29%) followed by a busy work schedule (7.71%) were the most common cause of poor medication adherence as reported by the respondents [Figure 1]”

Table 3

#5… I found here the problem you made in the abstract that confused the readers of your manuscript.

In the variable, “how many years ago were you first diagnosed as hypertensive (in years)” you used this category: < 5 years, < 10 years, and ≥ 10 years… but you just presented < 10 and > 10 in the abstract.

This category should be re-write because it is a bit confusing. You need categorize this as follows and please clarify the abstract section in this method. <5, 5-10, and ≥ 10 years.

In addition, please be careful where to place the boundaries in your case. For instance, is 5 included in the lower or in the middle?

Better to use this variable... “How many years ago were you first diagnosed as hypertensive (in years)…concisely and clearly like... “Duration of hypertension diagnosis or Duration since diagnosed for hypertension or Duration since first diagnosed as hypertensive”

#6…Better to incorporate the outcome variable (medication adherence) with frequency of good adherence and poor adherence for all independent variables. Then every reader easily understand particularly the odds ration from respective variables.

#7... Here also you didn’t want to mention about a p-value you were going to declare a statistical significant association. Even, you didn’t inform us in the statistical analysis section of the method. So how readers know about it whether it is < 0.05 or something different …?

Discussion

#1...Just you only mentioned that the overall prevalence of good adherence is higher than poor adherence and this is in line with some studies in Bangladesh [], Ethiopia [], and Malaysia [].

I don’t think this is enough. Is the proportion of good adherence being more than 50% by itself enough to say patients with hypertension had good medication adherence level? Of course, in your study, only around 54% had good medication adherence. You need to present in detail and compare with other evidence. Because you are talking about the prevalence, I expect you need to present and interpret your finding in terms of three points: In line [you mentioned], lower and, and higher findings as compared with your finding.

In general, the discussion should be organized by presenting the summarized findings in the first paragraph. Then discuss each finding of the study in terms of possible explanations for the disparity and consistence, your possible scientific suggestions, and its implications accordingly.

Limitation

The second limitation is somewhat unclear. Does it mean they were going to or unable to respond/participate because of their fear to their future treatment? Please could you clarify it more? In addition, please your limitation should be specific to your particular study.

Conclusion

No need of much talk here, and better to move some points to your discussion section. You should conclude here about your main outcome variables and their interpretation as I mentioned in the abstract section.

Reviewer #2: Thank you for the opportunity to review “Hypertension and Medication Adherence: An Institution-based Study at Hypertension and Research Center, Rangpur”

The manuscript is of interest to the reader. I commend the authors for nice work. The manuscript reads well. But the following points need to be addressed by authors.

Introduction

The introduction needs to be succinct and focused to hypertension and medication adherence.

NCDs is used for first time in the introduction… please mention in bracket before start using acronym.

What unique contribution this study would make if there were many studies on medication adherence in hypertension patients in Bangladesh. The authors need to assert the relevance of their study in the introduction.

Method

In the method and result section. The authors used two confusing categories of duration of hypertension diagnosis. Less than 10 years and less than 5 years are not mutually exclusive. Make sure you rewrite to ensure mutual exclusivity. This should be 5-10 year and <5 years. Otherwise, authors need to justify it for readers.

Despite the authors' claim of a larger sample size in the methods section, the sample size appears to be small, as evidenced by the very wide confidence intervals. The assertion of a large sample size in the methods section needs to be revised.

The authors stated that they included every variable in the regression. Including all uncorrelated variables in this small sample can lead to overfitting of the model. However, the actual number of included variables is not quite large. The authors better reflect what they actually did. It is good to be modest with their descriptions. Otherwise, a clear description of variable selection technique and inclusion criteria to the regression model need to be described.

The authors need to ensure they have obtained permission to use the MMAS-8 tool, as Morisky et al. typically require permission or payment for its use.

Result

Distribution of Medication Adherence and Table 2. I think simply expressing as chi squared test could be enough.

Table 3 and associated result description. The Hypertension duration categories need to be mutually exclusive.

Discussion

The authors' discussion is very shallow and should be made more relevant to their setting. The discussion section and conclusion are not seamlessly connected. The discussion's claim that good adherence is higher than poor adherence should not be the focus of the study. The fact that 45.17% of the sample exhibited poor adherence is significant and needs to be thoroughly discussed.

Limitation

“Firstly, due to the cross-sectional nature of the study, we could not establish any causal association with poor medication adherence among the participants.” It is a known generic limitation of cross-sectional studies. The authors better describe limitation only relevant to their specific study and setting.

Conclusion

“In conclusion, medication adherence is a critical aspect of hypertensive patient

management. Improving adherence could significantly enhance blood pressure control and reduce the risk of cardiovascular complications.” This is irrelevant. The authors should only conclude based on their result and specific to their setting/Bangladesh.

Minor points

Spellings errors and punctuation need to be addressed throughout.

6. PLOS authors have the option to publish the peer review history of their article (what does this mean? ). If published, this will include your full peer review and any attached files.

**Do you want your identity to be public for this peer review?** For information about this choice, including consent withdrawal, please see our Privacy Policy .

Reviewer #1: No

Reviewer #2: No

---

## [Author Response · Author response to Decision Letter 1]

28 Nov 2024

Reviewer #1:

Title: Hypertension and Medication Adherence: An Institution-based Study at Hypertension and Research Centre, Rangpur

Minor comments

Comment: The word/phrase “hypertension/hypertensive patients” is not a good usage of language because it focuses on labelling of patients. Despite you may find it in a lot of literature, but it has a negative and discriminative sense of language and not advised to use. “Patients with hypertension” is generally preferred in scientific writing. It focuses on the condition rather than labelling the patient. Please you need to change it in all sections of the manuscript.

Response: Thank you for pointing this out. We have made the necessary changes accordingly.

Comment: The MMAS-8 likely requires permission to use, right? "Have you obtained permission to use the MMAS-8 in your study? It's important to ensure you have the necessary permissions before publishing your findings. The authors need to ensure this. Other ways you are making yourselves and others busy with nothing.

Response: Thank you for your concern. Yes, we have obtained the permission to use the scale. We have now added the certificate number to the manuscript.

Comment: The journal requires a manuscript fully available without restriction, but you put restriction to your data availability. Please refer the Data availability policy of the journal.

Response: We have provided the data availability statement that is aligned with the journal’s requirements. Anyone can make a request to access the data from the NGHI repository using the email nghi.nsu@northsouth.edu.

Major comments

Title

Comment: The title as it stands is ambiguous, and it needs to be clarify with your interest to investigate. Determining the prevalence of medication adherence and its associated factors in patients with hypertension is the aim of the study you are going to investigate, right? So, please could you make it clear in this context?

Response: As per your suggestion, we have now modified the title of the study as follows:

Prevalence and Determinants of Medication Adherence among Hypertensive Patients: An Institution-Based Cross-Sectional Study

Abstract

Comment: Is Pearson’s chi-square test used to explore factors associated with medication adherence?

Response: Pearson’s chi-square test was applied to explore associations between categorical outcomes and explanatory variables. Necessary changes were made in the abstract and in the method sections of the manuscript.

Comment: bivariate and multivariable logistic regression analyses: is it bivariable or bivariate?

Response: Thank you for your concern. Although bivariate and bivariable analysis are used interchangeably in the literature, bivariable analysis is the appropriate word to use in this case. We have made this change in the abstract and method sections.

Comment: It would be good if you could include statistical analysis software and your declarations used for this particular study.

Response: We have stated the software, including the version, used for data analysis in this study as suggested (in the abstract and in the method section).

All statistical analyses were conducted using Stata version 17.0.

Comment: Result or results

Response: Supposed to be results, corrected!

Comment: “Duration of hypertension diagnosis (AOR: 0.32; 95% CI: 0.11-0.95 for hypertension diagnosed <10 years back and AOR: 0.28; 95% CI: 0.09-0.84 for >10 years)”. I didn’t find the point on these results, or if there is something not clearly presented, please could you clarify it? You mentioned both < 10 and > 10 years duration of hypertension diagnosis as a factor for medication adherence. What is the counterpart reference here? I mean, patients living < 10 and >10 years with hypertension had poor/good medication adherence than what?

Response: Thanks for pointing out this ambiguity. We have revised this portion of the result section to clarify our point, which reads as follows:

Among the study participants, married patients were found to have higher medicine adherence (AOR= 3.81; 95% CI: 1.34-10.89) than unmarried patients. Compared to patients who had hypertension for less than 6 years, patients suffering from hypertension for 6-10 years had 68% (AOR= 0.32; 95% CI: 0.11-0.95), and patients suffering for more than 10 years had 72% (AOR= 0.28; 95% CI: 0.09-0.84) lower odds of medicine adherence, respectively. Patients who were diagnosed by non-professionals had 81% (AOR= 0.19; 95% CI: 0.06-0.61) lower odds of medicine adherence compared to the patients who were diagnosed by health professionals.

Comment: Better to use “AOR =” instead of “AOR:”

Response: Thank you for your suggestion. We have made the necessary changes in the manuscript according to your suggestions.

Comment: Your conclusion is not aligned with your main outcome variable. Your main outcome variable is prevalence of medication adherence, right? If so, your conclusion should inform readers what does the outcome implies. These things mentioned here might be used for your justification and recommendation about the outcome.

Response: According to your suggestion, we have revised the conclusion section of the abstract, mainly focusing on the outcome of our research. Thank you!

This study observed low medication adherence among patients with hypertension, where forgetfulness and a busy work schedule were reported to be the primary reasons for such non-adherence. Patients who are not married, who have been suffering for a long, and who have not been diagnosed by a health professional manifested as significant influencing factors for non-adherence.

Introduction

Comment: organized and written well, but some evidences are mentioned without a citation, please you need to cite and acknowledge appropriate literature for it.

Response: Thank you for your valuable comment. Citations are now updated in the revised version of the manuscript.

Comment: there is a redundancy of the word hypertension or the hypertension in the first paragraph, better to re-write without redundant it.

Response: We have tried to eliminate the redundancy in the word ‘hypertension’ in the revised version of our manuscript.

Comment: Make sure of using updated evidence, while you are mentioning the WHO and other international organizations’ reports and evidence. For instance, the WHO published an update on 16 March 2023. https://www.who.int/news-room/fact-sheets/detail/hypertension#:~:text=Approximately%201%20in%205%20adults,33%25%20between%202010%20and%202030.

Response: We have updated the WHO citation in the revised manuscript.

Methods

Comment: “..Adult participants registered...”… What does registered means? Did they receive antihypertensive medications? If so, why we couldn’t mention them …adult participants who received antihypertensive medications?”

Response: Rangpur Hypertension Center is open to treat individuals with hypertension without needing to register for follow-up. Although the percentage of patients who opt out from registration is low, we wanted to exclude them from our study. We revised the sentence to make it clear. In addition, we also included the inclusion/exclusion criterion.

Comment: Why you chose six months? Is there any implication or rationale? Why not three months?

Response: Patients who get registered at the Rangpur Hypertension Center for follow-up receive frequent follow-ups (in-person and via telephone) for the first six months, and during that time, they are rigorously counseled about medication adherence. The frequency of follow-up then remains voluntary. Therefore, we wanted to recruit patients who went past that rigorous follow-up period.

Comment: You said participants were included based on the inclusion criteria. You should better to clarify these inclusion criteria more than this. You need to clarify our confusion in the abstract section above here.

Response: Thanks for your suggestion. We included a section in our revised manuscript stating the inclusion criteria, which clearly states the criteria for selecting the study participants.

Comment: What is 5th? You did not inform us how you did the sample size calculation, and you just wrote only “the sampling interval-K”. Please move this section including sampling technique you used after the sample size calculation because you are going to calculate k after determined the sample size.

Response: thank you for pointing this out. We have described the calculation of the sampling interval in the “study design, setting, and sampling” section of the manuscript.

Comment: Why you would like to use a larger sample size of 325, while you found a sample size of ~292 is enough for this study?

Response: We initially anticipated a sizeable non-response rate and wanted to approach more than 350 patients. Given their loyalty to the Hypertension center, all the 352 patients we approached agreed to participate. Therefore, we included all of them, given that they will contribute to obtaining a more precise estimate.

Comment: You used a thorough literature review to organize the questionnaire for this study. You need to cite and acknowledge these literatures used for developing your questionnaire.

Response: Thank you for your comment. We have acknowledged the literature that we used for the tool development in the “data collection” section of the manuscript and proper citations have been added.

Comment: Pearson’s chi-square test here in the method section is presented appropriately unlike in the abstract.

Response: Thank you for your comment. We have corrected it in the abstract as well.

Comment: bivariate, the same as state in the abstract

Response: As suggested, we have corrected it here and in the abstract section.

Comment: better to incorporate your declaration of statistical significant p-value here.

Response: Thank you for the comment. We have added a declaration for the p-value, which will be considered statistically significant in the “statistical analysis” section of our revised manuscript.

The P value less than 0.05 was considered statistically significant.

Results

Comment: When I just observed the number of respondents [352] in the table 1, I was surprised to observe a 100% response rate. This is rare in many studies. I wander if you could include this with some methods used to achieve a 100% response rate. Your experience may be valuable to other researchers.

Response: It came out as a complete surprise to us as well. We anticipate that selecting the registered patients of the hypertension center has contributed to that effect. Given that they received free treatment from the center, they may have been morally obligated to participate in the study, which made the response rate 100%.

Comment: Table 1 and its description: you described using texts almost all of the variables in the table 1. Please put only some texts to describe tables, other ways it looks duplication or no need of tables.

Response: Thank you for your comment. In the revised manuscript, we have highlighted only important demographics in the text and left the rest for the reader to explore from the table.

Comment: The same is true for Table 2 and its description texts. I think the need of table is to present variables in short, so focus on the main variable you would like to describe using texts and leave others in the table with reference [table citation] for the readers.

Response: Modified the description according to your suggestion highlighting only the interesting results.

Comment: Again you mentioned all variable using texts to figure 1, and then you are presenting as a figure again. Please avoid the figure and present only using texts or if you want to keep both mentioned the text like this: “Forgetfulness (20.29%) followed by a busy work schedule (7.71%) were the most common cause of poor medication adherence as reported by the respondents [Figure 1]”

Response: The revised version of the manuscript incorporates this suggested modification. Thank you!

Table 3

Comment: I found here the problem you made in the abstract that confused the readers of your manuscript. In the variable, “how many years ago were you first diagnosed as hypertensive (in years)” you used this category: < 5 years, < 10 years, and ≥ 10 years… but you just presented < 10 and > 10 in the abstract. This category should be re-write because it is a bit confusing. You need categorize this as follows and please clarify the abstract section in this method. <5, 5-10, and ≥ 10 years.

In addition, please be careful where to place the boundaries in your case. For instance, is 5 included in the lower or in the middle? Better to use this variable... “How many years ago were you first diagnosed as hypertensive (in years)…concisely and clearly like... “Duration of hypertension diagnosis or Duration since diagnosed for hypertension or Duration since first diagnosed as hypertensive”

Response: Thank you for pointing this out. The actual category for the variable “duration of hypertension diagnosis” was <6 years, 6-10 years, and >10 years. We have corrected the categories throughout the manuscript and revised the descriptions as suggested.

Comment: Better to incorporate the outcome variable (medication adherence) with frequency of good adherence and poor adherence for all independent variables. Then every reader easily understand particularly the odds ration from respective variables.

Response: Thank you for the comment. We have shown the frequency of the outcome variable in Table 2 for all the independent variables. Doing the same for Table 3, will it not be redundant? Although we intended to change Table 3, the other reviewer suggested excluding those frequencies even from Table 2. Therefore, we kept both Tables 2 & 3 unchanged so that both the reviewer's comments are satisfied.

Comment: Here also you didn’t want to mention about a p-value you were going to declare a statistical significant association. Even, you didn’t inform us in the statistical analysis section of the method. So how readers know about it whether it is < 0.05 or something different …?

Response: The revised version of the manuscript incorporated the p-value declaration in the “Statistical Analysis” section as suggested.

Discussion

Comment: Just you only mentioned that the overall prevalence of good adherence is higher than poor adherence and this is in line with some studies in Bangladesh [], Ethiopia [], and Malaysia []. I don’t think this is enough. Is the proportion of good adherence being more than 50% by itself enough to say patients with hypertension had good medication adherence level? Of course, in your study, only around 54% had good medication adherence. You need to present in detail and compare with other evidence. Because you are talking about the prevalence, I expect you need to present and interpret your finding in terms of three points: In line [you mentioned], lower and, and higher findings as compared with your finding. In general, the discussion should be organized by presenting the summarized findings in the first paragraph. Then discuss each finding of the study in terms of possible explanations for the disparity and consistence, your possible scientific suggestions, and its implications accordingly.

Response: Thank you for your valuable suggestion on how to improve the discussion section. In the first paragraph of the discussion section of the revised manuscript, we have added a paragraph summarizing the key findings of this study. We also compared and discussed the prevalence findings that were inline, lower, and higher from what we have observed.

Limitation

Comment: The second limitation is somewhat unclear. Does it mean they were going to or unable to respond/participate because of their fear to their future treatment? Please could you clarify it more? In addition, please your limitation should be specific to your particular study.

Response: We have rewritten the limitation section in the revised version of the manuscript to address your concern, and the content of the limitation has been made specific to this particular study.

Conclusion

Comment: No need of much talk here, and better to move some points to your discussion section. You should conclude here about your main outcome variables and their

---

## [Decision Letter · Decision Letter 1]

7 Mar 2025

Prevalence and Determinants of Medication Adherence among Hypertensive Patients: An Institution-Based Cross-Sectional Study

PONE-D-24-18421R1

Dear Dr. Sharif,

We’re pleased to inform you that your manuscript has been judged scientifically suitable for publication and will be formally accepted for publication once it meets all outstanding technical requirements.

Kind regards,

Dr Buna Bhandari

Academic Editor

PLOS ONE

Additional Editor Comments (optional):

Reviewers' comments:

Reviewer's Responses to Questions

**Comments to the Author**

1. If the authors have adequately addressed your comments raised in a previous round of review and you feel that this manuscript is now acceptable for publication, you may indicate that here to bypass the “Comments to the Author” section, enter your conflict of interest statement in the “Confidential to Editor” section, and submit your "Accept" recommendation.

Reviewer #2: All comments have been addressed

2. Is the manuscript technically sound, and do the data support the conclusions?

Reviewer #2: (No Response)

3. Has the statistical analysis been performed appropriately and rigorously? 

Reviewer #2: (No Response)

4. Have the authors made all data underlying the findings in their manuscript fully available?

Reviewer #2: Yes

5. Is the manuscript presented in an intelligible fashion and written in standard English?

Reviewer #2: (No Response)

6. Review Comments to the Author

Reviewer #2: (No Response)

7. PLOS authors have the option to publish the peer review history of their article (what does this mean? ). If published, this will include your full peer review and any attached files.

**Do you want your identity to be public for this peer review?** For information about this choice, including consent withdrawal, please see our Privacy Policy .

Reviewer #2: No

---

## [Editor Report · Acceptance letter]

PONE-D-24-18421R1

PLOS ONE

Dear Dr. Sharif,

I'm pleased to inform you that your manuscript has been deemed suitable for publication in PLOS ONE. Congratulations! Your manuscript is now being handed over to our production team.

Kind regards,

on behalf of

Dr. Buna Bhandari

Academic Editor

PLOS ONE